# THE COGNITIVE CAPABILITIES OF GENERATIVE AI: A COMPARATIVE ANALYSIS WITH HUMAN BENCHMARKS

## ABSTRACT

There is increasing interest in tracking the capabilities of general intelligence foundation models. This study benchmarks leading large language models and vision language models against human performance on the Wechsler Adult Intelligence Scale (WAIS-IV), a comprehensive, population-normed assessment of underlying human cognition and intellectual abilities, with a focus on the domains of Verbal Comprehension (VCI), Working Memory (WMI), and Perceptual Reasoning (PRI). Most models demonstrated exceptional capabilities in the storage, retrieval, and manipulation of tokens such as arbitrary sequences of letters and numbers, with performance on the Working Memory Index (WMI) greater or equal to the $99.5^{th}$ percentile when compared to human population normative ability. Performance on the Verbal Comprehension Index (VCI) which measures retrieval of acquired information, and linguistic understanding about the meaning of words and their relationships to each other, also demonstrated consistent performance at or above the $98^{th}$ percentile. Despite these broad strengths, we observed consistently poor performance on the Perceptual Reasoning Index (PRI; range 0.1-$10^{th}$ percentile) from multimodal models indicating profound inability to interpret and reason on visual information. Smaller and older model versions consistently performed worse, indicating that training data, parameter count, and advances in tuning are resulting in significant advances in cognitive ability.

## 1 INTRODUCTION

Generative artificial intelligence (GenAI) refers to a class of models capable of creating new content, whether it is text, images, music, or even code (Team et al., 2023; Wu et al., 2023; Gardner et al., 2023; Ouyang et al., 2023). Unlike traditional AI systems designed for specific tasks, these models learn the underlying patterns and structures within vast datasets and then use this knowledge to generate novel outputs that often mimic human creativity (Zhao et al., 2023; Nath et al., 2024). The excitement surrounding GenAI stems from its potential to revolutionize numerous fields by performing human-like cognitive functions (Wang et al., 2024; Zhuang et al., 2023; Zhang et al., 2024). This ability to understand, learn, adapt, and create in a way that mirrors our own thought processes opens doors to groundbreaking applications across industries like art, design, research, and communication (Ko et al., 2023; Lu et al., 2024; Ge et al., 2024; Yang et al., 2024). However, human cognition encompasses a vast array of specialized abilities in the processing, storage, interpretation, and generation of information across auditory and visual channels to accomplish these unique human capabilities (Cao et al., 2024; Subramonyam et al., 2023).

Recent advances in generative AI have been driven by large parameter models (e.g., the GPT (Achiam et al., 2023), Gemini (Team et al., 2023) families) trained with a broad range of textual content. Scaling experiments have shown that testing losses often follow power law relationships with data, model size (number of parameters) and compute (Kaplan et al., 2020). This progress has extended to multimodal models that

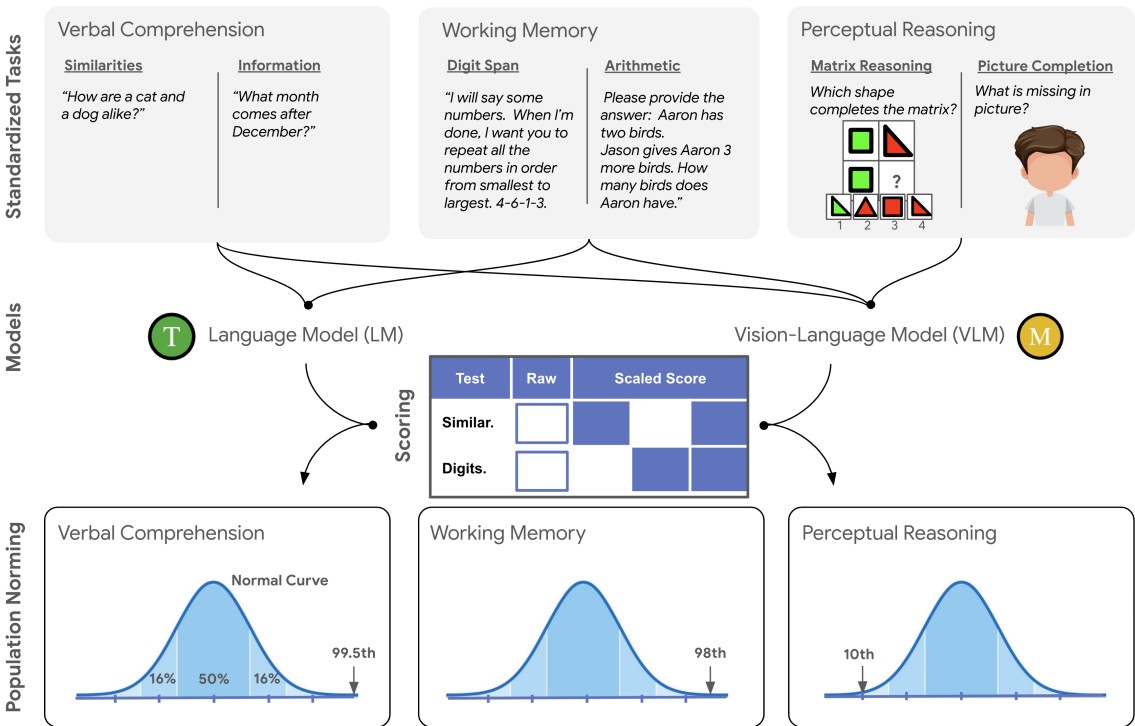

Figure 1: **Benchmarking Models Against Human Performance on the Wechsler Adult Intelligence Scale.** We perform a comprehensive, population-normed assessment of AI models against underlying human cognition and intellectual abilities, with a focus on the domains of Verbal Comprehension (VCI), Working Memory (WMI), andPerceptual Reasoning (PRI).

learn complex cross-modal relationships between representations such as language and visual media. The capabilities of these models has sparked debate about the potential impact of AI and proximity to artificial general intelligence (Bengio et al., 2024). The rapid progress increases the need for careful assessment of performance.

To assess their potential and limitations as models of human cognition, it is crucial to characterize the cognitive capabilities of generative models and understand how they compare directly to human ability (Ilić & Gignac, 2024). This will facilitate an understanding of pragmatic capabilities, limitations, and an ability to track progress just as is done with human normative cognitive tests of ability. It will also provide insight into unique streams of cognitive development of generative artificial intelligence under the assumption that models, like biological organisms, will develop unique capabilities based on their underlying neural architecture and environmental demands.

General cognitive capabilities have been systematically researched and benchmarked in humans for over a century (Spearman, 1961; Sternberg et al., 1981; Weiss et al., 2013). Refined standardized tests of ability and performance that are normed in large representative population-based samples allow for individual performance scores to be compared directly to normative performance. Because tests such as the Wechsler Adult Intelligence Scale–Fourth Edition (WAIS-IV; (Wechsler, 2008)) have population norms for comparison, individual scores can be used to make decisions about relative ability, and can be used to inform the allocation of special resources for education, housing, care, or competency. They can also be utilized to identify excep-

tional ability and gaps between areas of performance that are diagnostic of specific neurocognitive disorders (e.g., Autism Spectrum Disorder, Dyslexia; (Neisser et al., 1996)).

Beyond the focus on comparison to human performance, there may be value in the development of specific tests of GenAI cognition across discernible domains that are analogous to human tests akin to the development of analogous tests of cognition in animal models. Indeed, models of cognitive constructs such as working memory, semantic understanding, and visual processing, have facilitated research to understand the underlying neural architecture and mechanisms of cognition. Similar to the relative capabilities of the visual cortex of an hawk compared to a human – both display impressive but highly specialized abilities – we may find that GenAI models, by virtue of their architecture and context, develop novel patterns in cognitive functioning that prove to be quite different from humans, where, for example, consistent non-negligible positive correlations have been demonstrated between many cognitive ability test scores (Detterman & Daniel, 1989; Walker et al., 2023). This phenomenon, which is known as the Positive Manifold (Jensen, 1998), has been demonstrated to hold for LLMs using acceptable, yet non-validated, subtest proxies of the VCI and WMI cognitive ability tests (Ilić & Gignac, 2024).

We compared the performance of GenAI models on tests of cognitive performance that are commonly used to index human intellectual ability. We focused on tests of verbal ability including linguistic-based knowledge and conceptual understanding, capabilities to temporarily store, retrieve and manipulate arbitrary information, and capabilities to understand, reason, and manipulate visual information. Both individual test scores and composite index scores provide nuanced and interpretable information about intellectual capabilities and both were used to evaluate and better understand the capabilities of generative models.

## 2 METHODS

To facilitate the assessment of GenAI models using the Wechsler Adult Intelligence Scale, Fourth Edition (WAIS-IV), we implemented a series of methodological adaptations to accommodate the unique input and output modalities of these models. This involved converting traditional verbal and visual stimuli into text-based prompts (see Appendix B) and interpreting model-generated text outputs as responses to test items. Specific adaptations for each subtest, along with validation procedures, are detailed in 2.1. For this study, we selected a representative set of state-of-the-art (SOTA) large language models (LLMs) and vision language models (VLMs), encompassing a range of model sizes, architectures, and training datasets (See section 2.2).

### 2.1 TESTING MATERIALS, ADMINISTRATION, AND SCORING

Individual items from WAIS-IV subtests (Wechsler, 2008) were converted to prompts to be individually administered to the selected LLM (See Table 1 for a full list of tests with descriptions). Specifically, tasks comprising the verbal comprehension and working memory indices (VCI and WMI) were converted to language-based prompts and administered to all models including those with language-only capabilities as well as those with multimodal language and visual capabilities. Tests from the perceptual reasoning index (PRI) were only administered to multimodal models because they require both image recognition and language capabilities. We were unable to administer the Block Design subtest from the PRI, because it requires manual manipulation of real-world objects (cubes). However, valid PRI composite scores were still calculable substituting the alternate PRI subtest Figure Weights (Wechsler, 2008). Subtests from the Processing Speed Index (PSI) were not administered as there was no clear way to maintain fidelity to the WAIS-IV testing procedures that are required for valid comparison to human performance norms. A full-scale IQ score (FSIQ) could not be calculated for any model as PSI is a required component of FSIQ.

In each instance, the prompt presented to the model consisted of the instructions as outlined in the manual (Wechsler, 2008), including the example provided as part of the administration process. However, certain adaptations were necessary to accommodate the specific requirements of the model. Several tests involved

reading the test content, while others utilized visual and physical aids (e.g., cards and boards). These tests were converted into prompts, and in some cases, the translation provided the GenAI models with an advantage due to their ability to access the full context while generating responses. Additionally, phrases such as "You can only ask me to read the problem one more time" were omitted, as GenAI models are trained on instructions and often request repetition without responding to the actual assessment being conducted. Phrases such as *"Just say what I say"* or *"Listen"* were not removed and were left as they were originally presented, causing some models to mistakenly provide responses such as *"I am a text-based chat assistant and thus I cannot hear or repeat the numbers."* and *"Okay, I'm listening!."*. No further instructions or inquiries were provided. No additional instructions or queries were provided.

Each subtest consists of progressively challenging questions whose difficulty-level is normed against the population. For example, the Digit Span test, in which subjects are requested to verbally repeat a sequence of digits read aloud to them, begins with a sequence of three digits and progresses to a sequence of eight or nine digits depending on the subtest. Primary WAIS-IV composite indices and related cognitive domains are described below, along with brief descriptions of each subtest and administration procedures (See Table 1).

**Verbal Comprehension Index** These tests assess verbal knowledge and reasoning capacity. Subtests include: *Similarities*, in which the subject must state how two words are alike (e.g., "How are a kite and an airplane alike?"); *Vocabulary*, in which the subject provides definitions of words; *Information* in which the subject answers questions on a broad set of topics (e.g., "Who was the first president of the United States?"); and *Comprehension* in which subjects are asked to reason on common sense narrative scenarios.

**Working Memory Index** These subtests probe the capacity to store, manipulate, and reorder information, including reasoning numerically. *Digit Span* requires subjects to repeat back random number strings according to varying rules (i.e., Forwards, Backwards, and Sequencing conditions). *Arithmetic* requires subjects to solve math word problems of increasing difficulty without pen and paper. *Letter-Number Sequencing* requires subjects to manipulate arbitrary letter-number strings into alphabetic and numerical order.

**Perceptual Reasoning Index** Involves tests of visuospatial skills and nonverbal abstract reasoning. *Matrix Reasoning* (Ex: Fig. 2) requires subjects to choose the correct image from multiple options to successfully complete a visual pattern. *Visual Puzzles* (Ex: Fig. 3) requires subjects to select the correct tile to complete a picture, assessing mental rotation and spatial relations ability. In *Figure Weights* (Ex: Fig. 4), subjects are presented with an analogue scale depicting shapes of relative weight (e.g., two blue squares equals one red triangle). The subject is presented with a balanced scale demonstrating the weight relationship between objects, and an incomplete scale with shapes on only one side. They must then select from multiple choices to balance the incomplete scale. *Picture Completion* is a test in which subjects are presented with an image that is missing some key characteristic (e.g., an image of a woman without a shadow standing next to a sign that is casting a shadow). The subject must verbally indicate what is missing in the image.

All answers to prompts were scored by one of two clinical psychologists trained to administer and score the WAIS-IV. Any ambiguous results were reviewed by both psychologists to reach a consensus on scoring. Individual test scores were collated into raw index scores and then converted to age-normed scores with accompanying performance percentiles. As different norms exist for different age ranges, we selected the norms for age 25 years to 29 years and 11 months. In addition to the calculation of composite indices, we also compared individual subtests to domain-specific averages and composite scores in a series of discrepancy analyses, in order to probe the relative strengths and weaknesses of the models. The WAIS-IV provides standardized scoring for a variety of such comparisons, to determine if performance on an individual subtest might represent a statistically significant deviation from what would be expected in the population of those with similar performance profiles.

Table 1: **Summary of Tests.** Indices, subtests, and cognitive dimensions measured by the WAIS-IV.

| | Index / Subtest | Cognitive Dimensions | Administration | Assessed |
|---|---|---|---|---|
| **Verbal Comprehension** | Similarities | Verbal abstract reasoning, concept formation | Examinee verbally explains how two words are alike. | (T) (M) |
| | Vocabulary | Language comprehension, word knowledge | Examinee provides verbal definitions of words. | (T) (M) |
| | Information | General fund of knowledge, long-term memory recall | Examinee answers questions covering a broad range of general knowledge areas. | (T) (M) |
| | Comprehension | Social judgment, common sense, practical reasoning | Examinee answers questions about social situations and common sense problem-solving. | (T) (M) |
| **Perceptual Reasoning** | Block Design | Visual construction, spatial reasoning | Examinee manipulates blocks to reproduce visual designs. | N/A |
| | Matrix Reasoning | Nonverbal abstract reasoning, pattern recognition | Examinee selects the missing element that completes a visual pattern. | (M) |
| | Visual Puzzles | Mental Rotation, Spatial reasoning, part-whole analysis | Examinee chooses pieces that fit together to create a complete visual image. | (M) |
| | Figure Weights | Quantitative and analogical reasoning | Examinee views scales and selects the response option that balances the scale. | (M) |
| | Picture Completion | Visual perception, attention to detail | Examinee identifies the missing element from a picture. | (M) |
| **Working Mem.** | Digit Span | Attention, working memory, mental manipulation | Examinee recalls a series of digits forward and backward. | (T) (M) |
| | Arithmetic | Calculation Ability | Examinee mentally solves arithmetic word problems. | (T) (M) |
| | Letter-Num. Seq. | Working memory, alphanumeric sequencing | Examinee performs alphanumeric sequencing of letter-number strings. | (T) (M) |
| **Process. Speed** | Symbol Search | Visual scanning, target discrimination, processing speed | Examinee rapidly scans a search group and indicates whether a target symbol is present. | N/A |
| | Coding | Graphomotor processing speed | Examinee copies symbols paired with geometric shapes as quickly as possible. | N/A |
| | Cancellation | Selective attention, visual scanning | Examinee scans an array of visual stimuli and marks target items according to a rule. | N/A |

(T) = Text-Only Models, (M) = Multi-modal (Text and Image) Models

## 2.2 Models

Language only models included OpenAI's GPT-3.5 Turbo, Google's Gemini Nano, Gemini Pro, and Gemini Advanced. Multimodal models included OpenAI's GPT-4 Turbo, GPT-4o, Google's Gemini Flash, Gemini Goldfish and Anthropic's Claude 3 Opus, Claude 3.5 Sonnet. While model characteristics including the underlying training data, number of parameters, or internal tuning approach in the model are not publically available, key characteristics of the models are known that can be informative about factors that influence performance. First, we are able to compare progressive versioning of models to qualitatively compare performance. As an example, GPT-3.5 Turbo, Gemini Pro, and Claude 3 Opus are all earlier versions when compared to GPT-4 Turbo, GPT-4o, Gemini Advanced, and Claude 3.5 Sonnet. Further, Gemini Flash and Gemini Nano are intentionally designed as smaller models to be hosted on hardware chips rather than in cloud computing where the models can be much larger. Finally, Claude models indicate that they optimize

for advanced reasoning. While all of these pieces of information are limited due to the lack of public disclosure, they do allow us to determine if 1) larger parameter versus smaller parameter models perform better; 2) models intentionally optimized for reasoning do improve reasoning capabilities.

## 3 RESULTS

Index level results (VCI, WMI, PRI) demonstrate both exceptional capabilities and significant deficits when compared to a normative population (See Table 2 for full results). All but one model demonstrated abilities on the VCI in the Very Superior range, scoring within the top percentile of human normative ability with the exception of Gemini Flash which fell in the Superior range ($82^{nd}$%ile) and Gemini Nano ($23^{rd}$ %ile) which demonstrated ability in the Borderline range. The majority of models also performed in the Very Superior range on the WMI, with the exception of Gemini Nano ($37^{th}$%ile; Low Average). However, among the models with multimodal capabilities, performance on the PRI demonstrated a significant and consistent deficit with all models demonstrating capabilities in the Extremely Low range ($< 1^{st}$%ile) with the lone exception of Claude Sonnet which demonstrated ability in the Low Average range ($10^{th}$%tile).

Next, we analyzed individual differences in index-level results to understand relative strengths and weaknesses within domains of intellectual ability (See Table 3 for full Results). First, we observed a consistent relative strength in WMI compared to other indexes representing a statistically significant difference compared to the normative population ($p<.15$ or $p<.05$ level). Overall, this indicates that models' generally evidence stronger ability to process, store and manipulate information compared to any other ability, including language. Further, all models with visual/multimodal capability to perform tests comprising the PRI demonstrated significantly worse perceptual reasoning performance $p<.05$) when compared verbal comprehension and working memory indexes. This indicates a general relative weakness that is invariant across distinct developers and models in the ability to understand and reason on visual information.

Next, we examined performance on individual tests within each index (See Table 3 and Table 4). Within the WMI, a consistent relative weakness in Arithmetic, was observed in comparison to Digit Span, indicating a relative weakness in mathematical reasoning compared to the ability to encode, manipulate, and reorganize numeric values. Gemini Nano demonstrated further relative weaknesses in the ability to sequence and reverse numbers compared to the ability to encode and retrieve simple numeric lists. Analysis of the VCI demonstrated consistent relative strengths across models from independent developers, on the Information subtest compared to Similarities and Vocabulary ($p<.05$ and $p<.15$). This indicates that models are particularly strong in the storage and retrieval of natural language-encoded knowledge – often referred to as "crystallized knowledge" in human subjects – in comparison to the cognitive capabilities required for verbal analogic and verbal abstract reasoning and semantic understanding. This difference in relative performance persisted across generations of models indicating that models are consistently stronger in predicting the write answer than understanding and reasoning with language concepts. Finally, within the PRI, relative differences in opposite directions between best-in class models were observed with GPT-4o demonstrating stronger relative ability in decoding Visual Puzzles and Claude Sonnet demonstrating significantly stronger relative performance in Matrix Reasoning.

Wide variability in performance was observed on the VCI. While scores were highly skewed towards exceptional performance ($99.5^{th}$%ile to $>99.9^{th}$%ile, models that are known to be small parameter counts demonstrated the worst relative and absolute performance (Gemini Flash = $82^{nd}$%ile; Gemini Nano = $23^{rd}$%ile). Similarly, most models performed exceptionally on the WMI (Range, $95^{th}$ %ile - $>99.9^{th}$ %ile) with the exception of Gemini Nano ($37^{th}$ %ile). In contrast, all models tested demonstrated poor performance on the PRI (Range, $<.01^{th}$%ile - $10^{th}$%ile), with skewed performance such that the majority of models fell below the $2^{nd}$%ile. The exception is Claude Sonnet which demonstrates a significant increase over its predecessor Claude Opus indicating that while performance overall is poor, there is measurable progress in the ability for generative models to reason on visual concepts.

Table 2: **Model Scores on the WAIS-IV.** Composite index standard scores, individual subtest scaled scores, and percentile rankings for each generative AI model tested.

| | OpenAI | | | Google | | | | | Anthropic | |
|---|---|---|---|---|---|---|---|---|---|---|
| | **GPT-3.5 Turbo** | **GPT-4 Turbo** | **GPT-4o** | **Gemini Nano (1.0 S)** | **Gemini Pro 1.0 (1.0 M)** | **Gemini Adv. (1.0 XL)** | **Gemini Flash (1.5 S)** | **Gemini Goldfish (1.5 M)** | **Claude 3 Opus** | **Claude 3.5 Sonnet** |
| **WAIS-IV Composite Index** **WAIS-IV Subtests** | Scaled Score (%) | Scaled Score (%) | Scaled Score (%) | Scaled Score (%) | Scaled Score (%) | Scaled Score (%) | Scaled Score (%) | Scaled Score (%) | Scaled Score (%) | Scaled Score (%) |
| **Verbal Comprehension (VCI)** **Standard Score (%ile)** | **143 (99.8)** | **136 (99)** | **141 (99.7)** | **89 (23)** | **132 (98)** | **134 (99)** | **114 (82)** | **141 (99.7)** | **138 (99)** | **145 (99.9)** |
| Similarities | 13 (84) | 14 (91) | 15 (95) | 1 (0.1) | 8 (25) | 13 (84) | 14 (91) | 14 (91) | 13 (84) | 14 (91) |
| Vocabulary | 19 (99.9) | 15 (95) | 16 (98) | 4 (2) | 19 (99.9) | 15 (95) | 6 (9) | 17 (99) | 17 (99) | 19 (99.9) |
| Information | 19 (99.9) | 19 (99.9) | 19 (99.9) | 19 (99.9) | 19 (99.9) | 18 (99.6) | 19 (99.9) | 19 (99.9) | 19 (99.9) | 19 (99.9) |
| Comprehension | 19 (99.9) | 18 (99.6) | 19 (99.9) | 12 (75) | 18 (99.6) | 19 (99.9) | 19 (99.9) | 19 (99.9) | 17 (99) | 19 (99.9) |
| **Working Memory Index (WMI)** **Standard Score (%ile)** | **139 (99.5)** | **150 (>99.9)** | **150 (>99.9)** | **95 (37)** | **139 (99.5)** | **150 (>99.9)** | **139 (99.5)** | **145 (99.9)** | **150 (>99.9)** | **145 (99.9)** |
| Digit Span | 19 (99.9) | 19 (99.9) | 19 (99.9) | 11 (63) | 19 (99.9) | 19 (99.9) | 19 (99.9) | 19 (99.9) | 19 (99.9) | 19 (99.9) |
| DS Forward | 18 (99.6) | 18 (99.6) | 18 (99.6) | 18 (99.6) | 18 (99.6) | 18 (99.6) | 18 (99.6) | 18 (99.6) | 18 (99.6) | 18 (99.6) |
| DS Backwards | 18 (99.6) | 16 (98) | 19 (99.9) | 4 (2) | 19 (99.9) | 19 (99.9) | 19 (99.9) | 19 (99.9) | 19 (99.9) | 19 (99.9) |
| DS Sequencing | 19 (99.9) | 19 (99.9) | 19 (99.9) | 13 (84) | 19 (99.9) | 19 (99.9) | 18 (99.6) | 19 (99.9) | 19 (99.9) | 17 (99) |
| Arithmetic | 15 (95) | 19 (99.9) | 19 (99.9) | 7 (16) | 15 (95) | 19 (99.9) | 15 (95) | 17 (99) | 19 (99.9) | 17 (99) |
| LN Sequencing | 19 (99.9) | 19 (99.9) | 19 (99.9) | 1 (0.1) | 19 (99.9) | 19 (99.9) | 16 (98) | 19 (99.9) | 14 (91) | 16 (98) |
| **Perceptual Reasoning (PRI)** **Standard Score (%ile)** | – | **50 (<0.1)** | **58 (0.3)** | – | – | – | **50 (<0.1)** | **52 (<0.1)** | **50 (<0.1)** | **81 (10)** |
| Matrix Reasoning | – | 1 (0.1) | 1 (0.1) | – | – | – | 1 (0.1) | 1 (0.1) | 1 (0.1) | 8 (25) |
| Visual Puzzles | – | 1 (0.1) | 2 (0.4) | – | – | – | 1 (0.1) | 1 (0.1) | 0 (¡0.1) | 2 (0.4) |
| Figure Weights | – | 2 (0.4) | 6 (9) | – | – | – | 2 (0.4) | 4 (2) | 1 (0.1) | 10 (50) |
| Picture Completion | – | 1 (0.1) | 1 (0.1) | – | – | – | 1 (0.1) | 1 (0.1) | 1 (0.1) | 1 (0.1) |

Test level analyses demonstrate statistically significant relative strengths and weaknesses within individual composite indices. First, all models demonstrated perfect or near perfect scores on Information indicating that prior encoding and ability to access crystalized knowledge is exceptional compared to a normative population (Range, $99.6^{th}$%ile - $99.9^{th}$%ile). Next, within the VCI, multiple models demonstrated relative weaknesses in similarities (Gemini Nano, GPT 3.5, Gemini Pro; range indicating a relative difficulty reasoning on linguistic concepts compared to correctly retrieving crystallized linguistic knowledge. Gemini Nano and Gemini Flash both demonstrated relative weaknesses in vocabulary indicating these models struggle to understand language relative to other models. Gemini Nano also demonstrates lower scores in Comprehension when compared to Information, reinforcing that while this model has encoded and can retrieve crystallized knowledge well, it struggles to demonstrate human-level reasoning with linguistic information.

With regard to the WMI, models again performed exceptionally well with all demonstrating perfect scores in Digit Span with the exception of Gemini Nano which demonstrated a relatively poorer performance that was consistent with average human performance. Both Arithmetic and Letter Number Sequencing represented relative weaknesses for Gemini Nano. Further, when comparing subtests of Digit Span, we observed a relative weakness in Digit Span Backwards when compared to Digit Span Forward. Once again, Nano performed inconsistently, demonstrating abilities in digit span forward that were in the $99.6^{th}$%ile) while failing at manipulating similar numeric strings (Digit Span Backwards $2^{nd}$%ile). Further, Nano was unable to correctly perform any Letter-Number sequencing tasks (see Table 5 for complete results). Taken together, results indicate that Nano is proficient in encoding and retrieving the correct information but performs poorly when tasked with manipulating information, a key attribute of working memory.

Finally the PRI was assessed in the subset of multimodal models (GPT-4 Turbo, GPT-4o, Gemini Flash, Gemini Goldfish, Claude 3 Opus, Claude 3.5 Sonnet) demonstrating consistent performance in the Extremely Low range ( $0.2^{nd}$%ile). The best performing model was Claude 3.5 Sonnet which fell in the

Table 3: **Discrepancy Score Comparison Table for WAIS-IV Composite Indices.** Includes scale-level comparisons for Digit Span vs. Arithmetic and individual Digit Span component score comparisons.

| Index Level Comparisons | OpenAI | | | Google | | | | | Anthropic | |
|---|---|---|---|---|---|---|---|---|---|---|
| | GPT-3.5 Turbo | GPT-4 Turbo | GPT-4o | Gemini Nano (1.0 S) | Gemini Pro 1.0 (1.0 M) | Gemini Adv. (1.0 XL) | Gemini Flash (1.5 S) | Gemini Goldfish (1.5 M) | Claude 3 Opus | Claude 3.5 Sonnet |
| VCI - PRI (Difference) | – | 86** | 83** | – | – | – | 64** | 89** | 88** | 64** |
| VCI - PRI Base Rate (Critical Value, Significance) | – | 0.2% 8.32** | 0.2% 8.32** | – | – | – | 0.2% 8.32** | 0.2% 8.32** | 0.2% 8.32** | 0.2% 8.32** |
| VCI - WMI (Difference) | 4 | -14** | -9** | -6 | -7 | -16** | -25** | -4 | -12** | 0 |
| VCI - WMI Base Rate (Critical Value, Significance) | – | 14.1% 8.81** | 25% 8.81** | – | 29.7% 6.48* | 10.5% 8.81** | 3.0% 8.81** | – | 18.1% 8.81** | – |
| PRI - WMI (Difference) | – | -100** | -92** | – | – | – | -89** | -93** | -100** | -64** |
| PRI - WMI Base Rate (Critical Value, Significance) | – | 0.1% 8.81** | 0.1% 8.81** | – | – | – | 0.1% 8.81** | 0.1% 8.81** | 0.1% 8.81** | 0.1% 8.81** |
| **Scale Level Comparisons** | | | | | | | | | | |
| Digit Span Total - Arithmetic | 4** | 0 | 0 | 4** | 4** | 0 | 4** | 2* | 0 | 2** |
| Base Rate (Critical Value, Significance) | 9.4% 2.57** | – | – | 9.4% 2.57** | 9.4% 2.57** | – | 9.4% 2.57** | 29% 1.89* | – | 29% 1.89* |
| DS Forward-DS Backwards | 0 | 2 | -1 | 14** | -1 | -1 | -1 | -1 | -1 | -1 |
| Base Rate (Critical Value, Significance) | – | – | – | <0.10% 3.65** | – | – | – | – | – | – |
| DS Forward-DS Sequence | -1 | -1 | -1 | 5** | -1 | -1 | 0 | -1 | -1 | -1 |
| Base Rate (Critical Value, Significance) | – | – | – | 8.5% 3.60** | – | – | – | – | – | – |
| DS Backward-DS Sequence | -1 | -3* | 0 | -9** | 0 | 0 | 0 | 0 | 0 | 0 |
| Base Rate (Critical Value, Significance) | – | 17.3% 2.62* | – | 0.1% 3.56** | – | – | – | – | – | – |

Note: Base rates for Index Comparisons reflect cumulative percentages of the WAIS-IV normative sample (all ages) that obtained equivalent index score discrepancies. Base Rates for Scale Level comparisons reflect cumulative percentages of the entire WAIS-IV normative sample (i.e., full age range) that obtained equivalent discrepancy scaled scores. *$p < 0.15$; **$p < 0.05$

$10^{th}$%ile, consistent with Borderline performance in this domain. The Figure Weights test was a relative strength compared to other tests (Visual Puzzles, Matrix Reasoning) indicating a relative strength in analytical reasoning compared to pattern recognition or spatial reasoning. Of note, and despite overall poor performance, Claude 3.5 Sonnet demonstrated dramatic improvements over Claude 3 Opus in Matrix Reasoning and Figure Weights indicating that while performance in these domains lag behind language-based domains, models can be trained and optimized to acquire perceptual reasoning capabilities.

# 4    DISCUSSION

Results demonstrate that Generative AI models are capable of exceptional performance compared to normative human ability in key domains of cognition. First, and perhaps not surprising, all models demonstrate exceptional capabilities in storage, retrieval, and manipulation of arbitrary tokenized information such as sequences of numbers and letters while being significantly weaker in mathematical reasoning. While poor performance was most pronounced in small parameter models, the discrepancy between reasoning and

Table 4: **Verbal Comprehension vs Perceptual Reasoning.** Relative strengths and weaknesses for WAIS-IV verbal comprehension and perceptual reasoning subtests.

| Verbal Subtests | OpenAI | | | Google | | | | | Anthropic | |
|---|---|---|---|---|---|---|---|---|---|---|
| | GPT-3.5 Turbo | GPT-4o Turbo | GPT-4 | Gemini Nano (1.0 S) | Gemini Pro 1.0 (1.0 M) | Gemini Adv. (1.0 XL) | Gemini Flash (1.5 S) | Gemini Goldfish (1.5 M) | Claude 3 Opus | Claude 3.5 Sonnet |
| **Mean VCI Subtests** | 17.00 | 16.00 | 16.67 | 8.00 | 15.33 | 15.67 | 12.67 | 16.67 | 16.33 | 17.33 |
| Similarities (subtest - mean) | -4.00** | -2.00** | -1.67* | -7.00** | -7.33** | -1.67* | 0.33 | -2.67** | -3.33** | -3.33** |
| Base Rate (Critical Value, Significance) | 2% 1.91** | 15% 1.91** | 25% 1.56* | 1% 1.91** | 1% 1.91** | 25% 1.56* | – | 5% 1.91** | 2% 1.91** | 2% 1.91** |
| Vocabulary (subtest - mean) | 2.00** | -1.00 | -0.67 | -4.00** | 3.67** | -0.67 | -6.67** | 0.33 | 0.67 | 1.67** |
| Base Rate (Critical Value, Significance) | 10% 1.58** | – | – | 1% 1.58** | 1% 1.58** | – | 1% 1.58** | – | – | 15% 1.58** |
| Information (subtest - mean) | 2.00** | 3.00** | 2.33** | 11.00** | 3.67** | 2.33** | 6.33** | 2.33** | 2.67** | 1.67** |
| Base Rate (Critical Value, Significance) | 15% 1.64** | 5% 1.64** | 10% 1.64** | 1% 1.64** | 1% 1.64** | 10% 1.64** | 1% 1.64** | 10% 1.64** | 5% 1.64** | 25% 1.64** |
| **Perceptual Reasoning** | | | | | | | | | | |
| **Mean PRI Subtests** | – | 1.33 | 3.00 | – | – | – | 1.33 | 2.00 | 0.67 | 6.67 |
| Matrix Reas. (subtest-mean) | – | -0.33 | -2.00** | – | – | – | -0.33 | -1.00 | 0.33 | 1.33 |
| Base Rate (Critical Value, Significance) | – | – | 25% 1.92** | – | – | – | – | – | – | – |
| Visual Puzzles (subtest-mean) | – | -0.33 | -1.00 | – | – | – | -0.33 | -1.00 | -0.67 | -4.67** |
| Base Rate (Critical Value, Significance) | – | – | – | – | – | – | – | – | – | 1% 1.99** |

Note: Base rates for Verbal and Perceptual Reasoning subtests reflect cumulative percentages of the WAIS-IV normative sample (all ages) that obtained equivalent score discrepancies. $*p < 0.15$; $**p < 0.05$

information management persisted across model generations and developers indicating a generally stable discrepancy in ability.

Generative models also demonstrated exceptional verbal abilities, with variability in performance according to the size of the model. Once again, models were strongest in retrieval of stored information with consistent relative weaknesses in tasks that require understanding of linguistic concepts or the relationships between words and concepts. However, with the exception of Gemini Nano, all models demonstrated understanding as well as crystalized knowledge well above normative ability indicating that models generally excel in language-based tasks, even those requiring reasoning and understanding beyond simple regurgitation of acquired knowledge. In the case of small parameter models, they may be best for storing and retrieving information naturalistically (e.g., in the context of hardware constraints) but may not be capable of understanding or manipulating that information to solve a problem.

Finally, the dramatically poorer performance on visual processing tasks indicates that generative models, as they stand today, have profound deficits in the ability to understand the meaning or relationship in visual representations. Across developers and versions, models could not understand the meaning of objects, reason, problem-solve, or detect abnormal patterns in visual representations. There is some indication that current modeling approaches can lead to the acquisition of these abilities as Claude 3.5 Sonnet demonstrated profound increases over the previous generation (Claude 3 Opus). Claude 3.5 Sonnet showed advances over its predecessor in Matrix Reasoning ($0.1^{th}$%ile vs. $25^{th}$%ile), which measures the ability to detect meaningful patterns in visual stimuli and Figure Weights which indexes the ability to understand and reason on mathematical relationships that are visually presented ($0.1^{th}$%ile vs. $50^{th}$%ile). No such improvement was

Table 5: Longest Digit and Letter-Number Sequencing Spans with discrepancy analyses.

| | OpenAI | | | Google | | | | | Anthropic | |
| | GPT-3.5 Turbo | GPT-4 Turbo | GPT-4o | Gemini Nano (1.0 S) | Gemini Pro 1.0 (1.0 M) | Gemini Adv. (1.0 XL) | Gemini Flash (1.5 S) | Gemini Goldfish (1.5 M) | Claude 3 Opus | Claude 3.5 Sonnet |
|---|---|---|---|---|---|---|---|---|---|---|
| Longest Digit Span Forwards (LDSF) | 9 | 9 | 9 | 9 | 9 | 9 | 9 | 9 | 9 | 9 |
| LDSF Base Rate (25-29 yrs.) | (17.5%) | (17.5%) | (17.5%) | (17.5%) | (17.5%) | (17.5%) | (17.5%) | (17.5%) | (17.5%) | (17.5%) |
| Longest Digit Span Backwards (LDSB) | 8 | 7 | 8 | 3 | 8 | 8 | 8 | 8 | 8 | 8 |
| LDSB Base Rate (25-29 yrs.) | (7%) | (18.5%) | (7%) | (97.9%) | (7%) | (7%) | (7%) | (7%) | (7%) | (7%) |
| Longest Digit Span Sequencing (LDSS) | 9 | 9 | 9 | 9 | 9 | 9 | 9 | 9 | 9 | 9 |
| LDSS Base Rate (25-29 yrs.) | (2.5%) | (2.5%) | (2.5%) | (2.5%) | (2.5%) | (2.5%) | (2.5%) | (2.5%) | (2.5%) | (2.5%) |
| Longest Letter-Number Sequence (LLNS) | 8 | 8 | 8 | 0 | 8 | 8 | 8 | 8 | 8 | 8 |
| LLNS Base Rate (25-29 yrs.) | (6%) | (6%) | (6%) | (100%) | (6%) | (6%) | (6%) | (6%) | (6%) | (6%) |
| **Longest Digit Span Comparisons** | | | | | | | | | | |
| LDSF-LDSB (difference score) | 1 | 2 | 1 | 6 | 1 | 1 | 1 | 1 | 1 | 1 |
| Base Rate (All Ages) | (86%) | (64.5%) | (86%) | (1%) | (86%) | (86%) | (86%) | (86%) | (86%) | (86%) |
| LDSF-LDSS (difference score) | 0 | 0 | 0 | 0 | 0 | 0 | 0 | 0 | 0 | 0 |
| Base Rate (All Ages) | – | – | – | – | – | – | – | – | – | – |
| LDSB-LDSS (difference score) | -1 | -2 | -1 | -6 | -1 | -1 | -1 | -1 | -1 | -1 |
| Base Rate (All Ages) | (66.5%) | (40%) | (66.5%) | (<1%) | (66.5%) | (66.5%) | (66.5%) | (66.5%) | (66.5%) | (66.5%) |

Note: Base rates for Longest Digit Spans and L-N Sequencing Span raw scores reflect cumulative percentages from the WAIS-IV normative sample that obtained equivalent raw scores within the selected age bracket (25-29 years). Base Rates for Digit Span and LN Sequencing Span discrepancy scores reflect cumulative percentages from the entire WAIS-IV normative sample (i.e., all ages ranges) that obtained equivalent discrepancy scores.

observed in visual puzzle solving or image completion tasks that test the ability to understand and make sense of visual information.

Given the relative complexity of visual information processing compared to auditory processing in both humans and other vertebrates that rely on complex visual capabilities, the gap between visual and auditory cognitive capabilities in generative AI might not be addressed through incremental improvements in architecture or model training but may require separate specialized architecture for visual and auditory processing with enhanced interaction capabilities, as is the case with vertebrates.

The above results clearly demonstrate that for current SoTA models the Positive Manifold, which posits positive correlations between different cognitive capabilities, holds when VCI and WMI are considered (as demonstrated in Ilić & Gignac (2024)), and fails to hold for when including PRI as well.

The current work presents with the significant limitations that the model parameters themselves (underlying training data, parameter count, tuning approach) are all proprietary across all examples and thus not available for comparative analysis. While this does not limit the validity of the tests as these parameters are not known with human subjects either, it does limit the ability to draw definitive conclusions about factors that influence performance. This hampers scientific inquiry into GenAI models as model organisms to understand the acquisition of cognitive capabilities. The study is further limited by the inherently non-standard approach to WAIS-IV administration. While we attempted to copy key testing conditions and excluded tests that could not conform, there is an inherent limitation in the difference in testing setup from that which the scores were normed on. Despite these limitations, the current work represents the first of its kind approach to benchmark GenAI against human norms of intelligence. The results demonstrate the unique cognitive capabilities of current generative AI models as well as relative weaknesses, while providing a path to advance discrete domains of cognition that underlie wide sets of human cognitive abilities.

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

## A  BROADER IMPACT

The development of benchmarks help to assess the performance of foundation models. Capable models need to be developed responsibility and with attention to their strengths and flaws. Leveraging knowledge and tools from disciples such as clinical and neuro-psychology has allowed us to develop a set of grounded tasks that shed insight on the functioning of LLMs that are complementary to existing public benchmarks.

However, we must acknowledge that these tasks were designed specifically for evaluating human cognitive functioning and therefore extrapolation of the results to performance on more mundane, real-world tasks and conclusions that compare language model abilities to human cognitive functioning need to be treated with care.

# B   PROMPT EXAMPLES

---

**Similarities**

How are a[n] (blank) and a[n](blank) alike? Example: How are a cat and a dog alike? They are both pets.

---

**Digit Span Forward**

I will say some numbers. When I am done, repeat them in the same order just as I said them. Example: I say 1-3; You say 1-3

---

**Digit Span Backward**

I will say some numbers. When I am done, repeat them in the reverse order of how I said them. Example: I say 1-4; You say 4-1

---

**Sequencing**

I will say some numbers. When I am done, I want you to repeat all the numbers in order from smallest to largest. Example: I say 1-4-2; You say 1-2-4

---

**Matrix Reasoning**

Look at this picture and select which of the numbered options at the bottom should be placed in the square that currently has a question mark. You should select the option that works to complete the pattern both going up/down and going side to side.

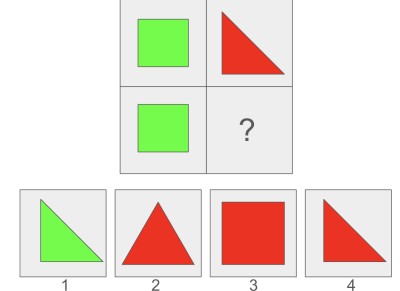

---

### Vocabulary

Listen carefully to each word that I say and tell me the meaning: Example: Pen: A writing utensil that uses ink.

### Arithmetic

I am going to read you some problems. Please provide the correct answer: Example: Aaron has two birds. Jason gives Aaron 3 more birds for his birthday. How many birds does Aaron now have. Correct Answer: 5

### Visual Puzzles

I want you to image that the picture at the top is a puzzle. Select the pieces that go together to make the puzzle. The pieces can not overlap with each other.

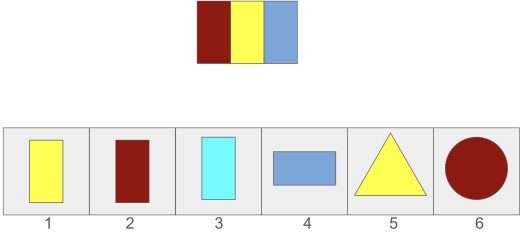

### Information

Now I am going to ask you some questions. Provide the correct answer. Example: What month comes after December? Answer: January

### Letter-Number Sequencing

I am going to say a series of letters and numbers. I want you to repeat them back in order starting with numbers then letters. Numbers should be listed from smallest to largest. Letters should be listed in alphabetical order. Example: 2-B-1-N-3-G; Answer: 1,2,3,B,G,N

### Figure Weights

Your goal is to place the item(s) that balance the scales Example: Look at this scale. It is not balanced. Choose the shape that will balance the scales.

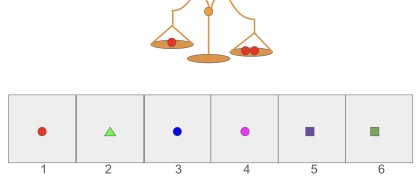

**Comprehension**

I'm going to ask some questions. Provide the best answer. Example: Why do people bathe? Answer: To clean themselves.

**Picture Completion**

I will show you some pictures that have something missing. Name what is missing from the picture. Example: Look at this example. What is missing from the picture?

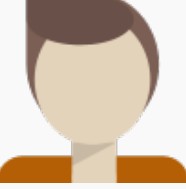

**Note:** Test is slightly altered to protect copy-written materials.

## C  VISUAL TASKS EXAMPLES

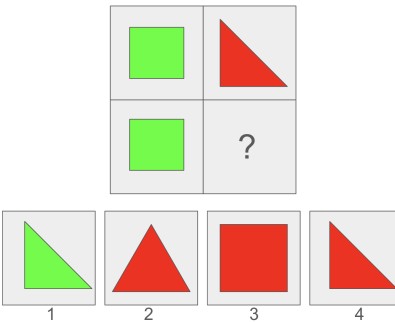

Figure 2: Example of the Matrix Reasoning test in which subjects are requested to identify a pattern between different rows and columns of the matrix. In this example, the first row is composed of green square, followed by a red triangle (from left to right). The second row also starts with a green square meaning that the correct option in the question mark is a red triangle. Answer: 2

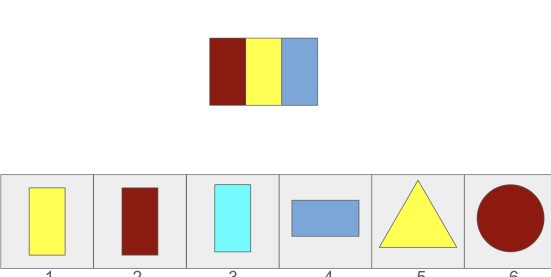

Figure 3: Example of the Visual Puzzles test in which subjects are requested to identify all the pieces that compose the provided design. In this example, the provided design is a three color rectangle. It can be composed by arranging rectangle (2), rectangle (1) and flipping rectangle (4), from left to right. Answer: 1,2,4.

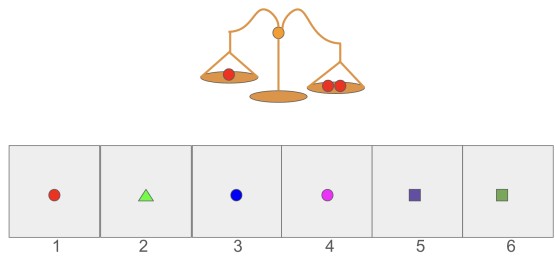

Figure 4: Example of the Figure Weights test in which subjects are requested to determine the correct way to balance the scales by finding the missing pieces. In this example, the scale on the left holds a single red circle, while the scale on the right holds two red circles. Considering that each red circle carries the same weight, attaining balance necessitates the addition of one red circle to the scale on the left. Answer: 1.

