# OpenReview forum: "The Cognitive Capabilities of Generative AI: A Comparative Analysis with Human Benchmarks"
_ICLR.cc/2025/Conference — Submitted to ICLR 2025_

### Official Review · Reviewer_oMqA · 2024-10-31

**Soundness:** 3
**Presentation:** 3
**Contribution:** 1
**Rating:** 5
**Confidence:** 4

**Summary:**

The paper attempts to measure the cognitive abilities of various large language models (LLMs) and vision-language models (VLMs) using a modified version of the WAIS-IV, a standardized intelligence test for humans. The authors find that while models perform exceptionally well on tasks involving verbal comprehension and working memory, they exhibit significant deficits in perceptual reasoning.

**Strengths:**

- The use of the WAIS-IV provides a familiar and well-established framework for comparing AI capabilities to human performance. This offers a more relatable and interpretable assessment compared to typical machine learning metrics.

- The methodology is robust and well presented.

**Weaknesses:**

- The main premise of the work is centred around testing LLMs using traditional IQ measures like WAIS test and there are several works which do similar analysis:
https://www.techrxiv.org/doi/full/10.36227/techrxiv.171778886.6283965 https://www.techrxiv.org/doi/full/10.36227/techrxiv.171778886.62839657
https://journals.plos.org/plosone/article?id=10.1371/journal.pone.0307097
https://www.techrxiv.org/doi/full/10.36227/techrxiv.22645561.v1
https://www.researchgate.net/profile/GuozhengRao/publication/334844361_How_Well_Do_Machines_Perform_on_IQ_tests_a_Comparison_Study_on_a_Large-Scale_Dataset/links/5dd279b94585156b351c1774/How-Well-Do-Machines-Perform-on-IQ-tests-a-Comparison-Study-on-a-Large-Scale-Dataset.pdf
Such works should be referred/ discussed and the novelty of the current work should be highlighted.

- One of the central findings of the paper that LLMs do well in language-based tasks while struggling with visual reasoning - is already well-established in the field.  For instance, Cao et al. (2024) directly address the "visual cognition gap" between humans and multimodal LLMs, a key finding echoed in this paper (https://arxiv.org/abs/2406.10424 )

- There also exists work on analysing working memory capacity of LLMs (https://ojs.aaai.org/index.php/AAAI/article/view/28868) and more discussion needs to be done to link to such works.

- The paper mostly uses all closed source models and it might be more interesting to use some open source models (Llama, Mistral etc) across different sizes for better understanding how trends of performance change.

**Questions:**

- Can you better characterize novelty of your work in light of similar existing works?

---

### Official Review · Reviewer_Kw3i · 2024-11-01

**Soundness:** 3
**Presentation:** 3
**Contribution:** 1
**Rating:** 3
**Confidence:** 3

**Summary:**

This study performs a benchmark evaluation on the cognitive abilities of state-of-the-art language models and vision language models. It uses the Wechsler Adult Intelligence Scale (WAIS-IV) which is a test developed for humans that incorporates several aspects of intelligence, such as working memory, perceptual capabilities, and verbal comprehension. The findings include that the models perform well on working memory tasks—greater or equal to the 99.5th percentile of human performance. The performance was also high on verbal comprehension, but surprisingly low on perceptual reasoning.

**Strengths:**

* The study addresses a question that is of broad interest, namely, how state-of-the-art models compare in performance to humans on cognitive tasks.
* The evaluation is original in the sense that it is grounded in an actual IQ test that is used on humans and it is performed by trained psychologists.
* It is interesting and noteworthy that the models perform so poorly on the perceptual reasoning tasks.
* The paper is clearly written and easy to follow.

**Weaknesses:**

While this is an overall sound study of how state-of-the-art generative models compare against humans on intelligence tests, I am unsure of its value as a machine-learning paper. From what I can assess, it does not contribute any new methodology, evaluation framework, or insights that could significantly advance the state of knowledge in the field. It is definitely interesting to benchmark these models against human performance, but there exists much literature on this topic already (little of which is cited here). See, e.g., Mahowald et al. (2023) for a review. It is thus unclear what this paper contributes beyond what is already in the literature. Moreover, existing work I am aware of tends to focus on specific linguistic or cognitive phenomena, which gives more concrete insights into where models may or may not fail. These things are important to study so that we can develop better evaluations, understand how LLMs fare as cognitive models in various regards, possibly inject inductive biases etc. The approach here, on the other hand, is more general. That makes it difficult to understand the implications of the results, especially for someone unfamiliar with the WAIS-IV framework.

Another weakness I see is that several decisions are made without rationales or citations. For example l145 “as GenAI models … often request repetition without responding to the actual assessment being conducted” or l182 “we selected the norms for age 25 years to 29 years and 11 month”.

In addition, the present study only uses closed-source models, with some details missing on the experimental setup, like the exact model snapshot used and the decoding method.

----

Kyle Mahowald et al. 2023. Dissociating language and thought in large language models.

**Questions:**

What do you see as the main contribution to the machine learning community? Could this work have implications for how we perform evaluations or how we can develop new models?

How large is the dataset? You mention that all answers were scored by clinical psychologists, so I would assume that it must be quite small to be feasible.

---

### Official Review · Reviewer_hNsV · 2024-11-01

**Soundness:** 2
**Presentation:** 3
**Contribution:** 2
**Rating:** 3
**Confidence:** 4

**Summary:**

This paper tests several generative AI models on different parts of the Wechsler Adult Intelligence Scale test, including verbal comprehension, working memory, and perceptual reasoning.  The top models excel on verbal comprehension and working memory tests but perform much worse on perceptual reasoning tests.

**Strengths:**

The paper is well-written and the experiments are explained clearly.

**Weaknesses:**

The paper lacks critical discussion regarding using tests designed to test human intelligence on LLMs.  The only such discussion is in the appendix (in the "Broader Impacts" section).  I think that discussion needs to be at the very beginning, and the authors need to argue why this kind of evaluation for an LLM gives useful information about its more general capacities.   Given the many problems that have been identified with LLM benchmarking using tests designed for humans, what can we learn from this particular evaluation? As one part of this discussion, the authors should address the question of whether the test items they presented to LLMs are in those LLMs pre-training data.

I was particularly confused by the "working memory" tasks.  For humans, these are given in an audio format, since it would not make sense to give such tests to them in a written format that they have access to while answering, since that would not test working memory.   But it seems here that the LLMs get these tests in their context window, and have access to them while answering?  I didn't understand what this is a test of, with respect to LLMs.

At the end, the authors state that their results "demonstrate that Generative AI models are capable of exceptional performance compared to normative human ability in key domains of cognition."   This seems like a fallacy to me, since there is not yet evidence that performance on these tests for LLMs is indicative of broader performance in "key domains of cognition".

The paper lacks citations to other work that evaluates LLMs on IQ tests, e.g.,

Wasilewski & Jablonski: Measuring the Perceived IQ of Multimodal Large Language Models Using Standardized IQ Tests

Huang & Li: Measuring the IQ of Mainstream Large Language Models in Chinese Using the Wechsler Adult Intelligence Scale

King: Administration of the text-based portions of a general IQ test to five different large language models

Lenhard & Lenhard: Beyond Human Boundaries: Exploring the Proficiency of AI Technology and its Potential in Psychometric Test Construction

(And there are probably other such papers in the recent literature.)

The paper also lacks any reference to critical discussion of giving tests designed for humans to LLMs, e.g.,

https://www.aisnakeoil.com/p/gpt-4-and-professional-benchmarks

https://www.science.org/doi/10.1126/science.adj5957

**Questions:**

How do you know the test items (or very similar items) aren't in the models' pre-training data?

"predicting the write answer than understanding and reasoning" -- write --> right

"the gap between visual and auditory cognitive capabilities in generative AI" -- do you mean language instead of auditory?

---

### Official Review · Reviewer_xECw · 2024-11-04

**Soundness:** 3
**Presentation:** 3
**Contribution:** 2
**Rating:** 5
**Confidence:** 3

**Summary:**

In the paper, the authors test current LLMs by OpenAI, Google, and Anthropic for cognitive capabilities according to the WAIS benchmark. The study is properly done and the paper is written well.

However, the current version of the paper provides too little novelty and significance in insight to warrant a publication in ICLR.

**Strengths:**

properly done study, well written paper

**Weaknesses:**

The numerical results for storage, retrieval, and manipulation are indeed not surprising and logical regarding the hardware capabilities pattern completion capabilities and vast representation space of the LLMs. Here, the paper could have made a difference in 1) identifying details through quantitative analysis, and 2) testing with other, more recent benchmarks on intelligence.

Furthermore, shortcomings of LLMs in visual processing/reasoning are not surprising either, as the tested models were either explicitly trained on uni-modal (text data) only, or we have no insight into training (but reasons to believe that). Here, the paper could have contributed in 1) testing  Multimodal LLMs as well, and 2) comparing between the tested LLMs to identify where qualitative differences lay in detail.

**Questions:**

none

---

### Meta-Review · Area_Chair_ksaa · 2024-12-20

**Metareview:**

This paper benchmarks modern language and vision-language models against human performance using the WAIS-IV intelligence test framework. While the study is well-executed and clearly presented, all reviewers identified significant limitations in novelty and contribution. The findings that models excel at verbal/memory tasks but struggle with visual reasoning largely confirm known capabilities. Critical weaknesses include: insufficient discussion of using human-designed tests for AI evaluation, limited citation of similar prior work, lack of open-source model comparisons, and absence of methodological innovations. The study provides incremental insights without advancing the field's understanding of AI capabilities or evaluation methods. Given these limitations and the authors' non-response during the rebuttal period, the paper falls below the acceptance threshold.

**Additional Comments On Reviewer Discussion:**

There was no author response or reviewer discussion during the rebuttal period. The initial reviews raised significant concerns about the paper's contribution, methodology, and novelty that remain unaddressed. The lack of author engagement with the review feedback further supports the decision to reject the submission.

---

### Decision · Program_Chairs · 2025-01-22

Reject